# Adherence to the Mediterranean Diet and Ultra-Processed Foods Consumption in a Group of Italian Patients with Celiac Disease

**DOI:** 10.3390/nu15040938

**Published:** 2023-02-13

**Authors:** Marta Tristan Asensi, Giuditta Pagliai, Sofia Lotti, Abigail Corrao, Barbara Colombini, Ilaria Giangrandi, Francesco Sofi, Monica Dinu

**Affiliations:** 1Department of Experimental and Clinical Medicine, University of Florence, 50134 Florence, Italy; 2Department of Human Science, Georgetown University, Washington, DC 20057, USA; 3Unit of Clinical Nutrition, Careggi University Hospital, 50134 Florence, Italy

**Keywords:** celiac disease, ultra-processed foods, NOVA classification, NFFQ, Mediterranean diet, Medi-Lite, gluten-free diet

## Abstract

Evidence on the consumption of ultra-processed foods (UPF) in adults with celiac disease (CD) and its impact on Mediterranean Diet (MD) adherence is still limited. Our aim was to determine UPF consumption and its relationship with MD adherence in a group of adults, according to the presence of CD. This case-control study included 103 adults with CD and 312 without CD. UPF intake was assessed using the NOVA Food Frequency Questionnaire (NFFQ), while MD adherence was assessed using the Medi-Lite score. UPF represented 14.5% of the diet of participants with CD (246 g/day) and came mainly from cereals-based products (29%) and sweets (24.2%). UPF consumption did not differ with the presence of CD, but participants with CD had significantly (*p* < 0.05) higher consumption of precooked pasta and pre-packaged breads. Participants with CD also reported a significantly lower MD adherence than participants without CD (9.4 vs. 10.4), with higher intake of meat and dairy products, and lower consumption of vegetables and fish. An inverse trend was found between UPF consumption and MD adherence in adults with CD, although not statistically significant. These findings highlight the importance of improving nutrition education for subjects with CD, which should not only focus on gluten exclusion.

## 1. Introduction

Celiac disease (CD) is a chronic autoimmune disease that affects the small intestine in genetically susceptible individuals. This condition is triggered by exposure to gluten, a protein complex found in wheat, rye, and barley [1]. Although the prevalence of CD is highest in childhood, being one of the most common chronic diseases in children, it is well known that it affects all age groups [1,2]. Moreover, CD is more prevalent in females than in males [2]. The most common presenting symptoms are malabsorption, diarrhea, steatorrhea, and weight loss resulting from damage to the intestinal mucosa. Other less common symptoms include abdominal pain, reflux esophagitis, or depression [1].

The mainstay of treatment for CD is strict adherence to a gluten-free diet [3]. Although the only restriction of the diet is the exclusion of gluten, the nutritional adequacy of this diet remains controversial, as some evidence suggests that the gluten-free diet is unbalanced due to lower fiber intake and greater intakes of sugar, total fat, and saturated fat [4]. A potential cause has been attributed to the quality of gluten-free products, which are generally associated with processed foods with low nutritional density [5,6,7]. Indeed, the so-called ultra-processed foods (UPF), known as “industrial formulations typically with five or more and usually many ingredients” [8], are associated with unbalanced diets due to their formulation rich in free sugars, fats, and salt [9,10,11].

In addition, some studies also suggest that people with CD do not adequately consume the different food groups that are part of a healthy diet [12]. One of the dietary patterns with robust evidence of its beneficial effect on health status is the Mediterranean diet (MD) [13]. To date, several studies in Mediterranean countries have attempted to investigate whether people with CD follow the MD, particularly in the pediatric population [12,14,15,16]. However, scientific evidence of UPF consumption in adults with CD and the possible impact it may have on MD adherence is still limited. For this reason, the aim of the present study was to determine the consumption of UPF and its possible relationship with MD adherence in a group of adults with CD compared to a group of adults without this condition.

## 2. Materials and Methods

### 2.1. Study Design and Data Collection

All study participants were recruited sequentially from patients referred to the Clinical Nutrition Unit of Azienda Ospedaliera Universitaria Careggi, Florence, Italy, during the period from January to December 2021. The cases consisted of all patients with CD who made a first nutritional visit to our unit during the study period. The controls were also patients who had a first nutritional visit at our unit, but without a diagnosis of CD or other disease. For optimal comparison, each case was matched with 3 controls, with age and gender matching. All subjects who were asked to participate in the study agreed. Data were collected through a self-administered questionnaire created with the online survey platform SurveyMonkey (www.surveymonkey.com, accessed 31 September 2022) [17]. Prior to starting the questionnaire, participants were asked to read the study project information sheet and to sign the informed consent form. Afterwards, participants were asked to fill a brief questionnaire on sociodemographic characteristics (age, gender, weight, height, civil status, and educational level) and two validated questionnaires to collect data on adherence to MD and UPF intake. Body mass index (BMI) was calculated as weight (kg)/height (m^2^). Civil status was classified as single, married/partnered, and divorced/widowed, while education level was classified as secondary school, high school, and university. This observational study was conducted in conformity with the guidelines set out in the Declaration of Helsinki and was approved by the Ethics Committee of the Tuscany Region, Azienda Ospedaliera Universitaria Careggi, Florence, Italy [CEAVC 18353/OSS].

### 2.2. NOVA Food Frequency Questionnaire (NFFQ)

Food consumption was assessed using the NOVA Food Frequency Questionnaire (NFFQ), a validated questionnaire aimed to evaluate the food consumption of Italian adults according to NOVA group classification [18]. Participants were asked to fill the 94 items of the NFFQ by indicating their typical frequency of consumption and portion size in their diet for a typical month within the last 12 months. Participants could choose 1 of 10 consumption frequency options, ranging from never or less than once a month to daily consumption. Portion sizes included 6 options, ranging from 0.5 to 3 portions, identified according to Italian reference portions and the portions indicated on food labels. The 94 items are subdivided into 9 main food groups: (1) fruits and nuts; (2) vegetables and legumes; (3) cereals and tubers; (4) meat and fish; (5) milk, dairy products, and eggs; (6) oils, fats, and seasonings; (7) sweets and sweeteners; (8) beverages; and (9) other. The amount consumed for each item, and the larger food group, was calculated in grams per week and grams per day.

All food and beverages included in the NFFQ were categorized according to the level of food processing, as unprocessed and minimally processed (MPF); processed culinary ingredients (PCI); processed (PF); and UPF. For the analysis of NFFQ data, PCI products were grouped with PF, as they are to be consumed with other foods, not alone as an independent food group. The weight ratio of foods within each level of processing was calculated, rather than the energy ratio, to more effectively represent processed foods that do not provide calories and non-nutritional components of processed foods.

### 2.3. The Medi-Lite Adherence Score

Participants also completed the Medi-Lite score [19], an evidence-based tool developed in 2014 and validated in 2017 [20], to determine individual adherence to MD. This questionnaire assigns points from 0 to 2 to daily and/or weekly consumption of 9 food groups, in accordance with the MD, to generate a final score ranging from 0 (lowest adherence to MD) to 18 (highest adherence to MD). Point values are assigned according to the frequency of consumption, using reference portions; the quantities chosen as cut-offs for each item were calculated on the basis of available literature linking the consumption of typical and non-typical Mediterranean foods to health indicators [19]. For foods typical of the MD, including fruits, vegetables, cereals, legumes, and fish, 2 points are assigned to the highest level of consumption, 1 point to the intermediate level, and 0 points to the lowest level of intake. Similarly for olive oil, 2 points are awarded for regular use, 1 for frequent use, and 0 for occasional use. Further, 2 points are assigned to the lowest intake level, 1 to the intermediate level, and 0 to the highest level of consumption for meat and meat products and dairy products, which are foods non-typical of the MD. For alcohol, 2 points are given to the intermediate level of consumption, 1 for the lowest, and 0 for the highest intake level.

In this study, the optimal intake for individual food groups was defined as the choice that produced > 2 points and corresponded to the following consumption levels: fruit > 2 portions per day, vegetables > 2.5 portions per day, cereals > 1.5 portions per day, dairy products < 1 portion per day, meat and meat products < 1 portion per day, fish > 2.5 portions per week, legumes > 2 portions per week, alcohol 1–2 alcohol units per day, and regular use of olive oil.

### 2.4. Statistical Analysis

Statistical analysis was performed using the statistical package IBM SPSS Statistics for Macintosh, version 28.0 (IBM Corp., Armonk, N.Y., USA). Continuous variables are expressed as mean ± standard deviation (SD) or median and range (min–max), as appropriate. Categorical variables are reported as frequencies and percentages. The Mann–Whitney test was used to compare participants with CD and participants without CD, while the Chi-square test was used to test proportions.

A general linear model adjusted for age, gender, BMI, education level, civil status, and daily food consumed was conducted to assess daily UPF consumption, according to the presence of CD. Since this test assumes a normal distribution of data, non-normally distributed data were transformed into logs, and further analyses were performed with the processed data. However, to facilitate interpretation, the log data were again converted to the original scale (antilog) and presented as geometric means with 95% confidence intervals (CIs). To assess the possible relationship between MD adherence and UPF consumption in participants with CD and participants without CD, they were divided into tertiles (1st tertile ≤ 10%; 2nd tertile = 10–17%; 3rd tertile ≥ 17%) according to their level of UPF intake (proportion by weight). A *p*-value < 0.05 was considered statistically significant.

## 3. Results

From January to December 2021, a total of 112 adults with CD were enrolled in the study as cases and matched with 336 adults without CD as controls. Subjects who did not answer all test questions were excluded, leaving a total of 103 cases with CD (92%) and 312 controls without CD (93%), for a total of 415 participants. The mean age of study participants was 39.6 ± 13.1 years, 84.3% of whom were women. About 25% of the subjects was overweight or obese (mean BMI 23.1 ± 4.2 kg/m^2^). Participants with CD had an average of 11.4 ± 8.6 years of CD progression since diagnosis. The sociodemographic characteristics of the subjects according to the presence of CD are shown in Table 1. More than half of the study population reported being married or with a partner. Overall, the study population was highly educated; a significant difference was found amongst participants with CD reporting secondary school as the highest educational level achieved as compared to participants without CD. No other significant differences between the two groups were observed.

### 3.1. Ultra-Processed Foods’ Consumption

UPF represented 14.5 ± 8% of the diet of participants with CD, which was equivalent to 246.3 ± 139.2 g/day. These values did not differ significantly from those reported in participants without CD, which showed a dietary percentage of 15.7 ± 9.5% of UPF equivalent to 264.4 ± 163.5 g/day. Table 2 reports the consumption of single UPF categories in the two groups. After adjustment for possible confounding factors, such as gender, age, and total food consumed (g/day), significant differences between participants with CD and without CD were found in the consumption of vegetables and legumes (*p* = 0.021) and meat and fish (*p* = 0.016), with lower consumption in participants without CD compared to participants with CD (−47.2% and −28.7%, respectively). Although no significant differences in overall consumption of ultra-processed cereals and tubers were found between the two groups, participants with CD were significantly more likely to consume ready-to-heat pasta (*p* = 0.033) and pre-packaged breads and bread alternatives (*p* = 0.012) than participants without CD (+32% and+25.5%, respectively). No significant differences were found in the consumption of the other categories of UPF between the two groups.

The food categories that contribute the most to UPF consumption in participants with CD were cereals and tubers (29%), sweets and sweeteners (24.2%), and beverages (12.7%).

### 3.2. Adherence to the Mediterranean Diet

Participants with CD reported a significantly (*p* < 0.001) lower adherence score to the MD (9.4 ± 2.2) than participants without CD (10.4 ± 2.5). Figure 1 shows the results obtained for the different food groups considered in the Medi-Lite questionnaire, according to whether the optimal recommendations of the MD were met. A significantly larger number of participants with CD did not meet the recommended portions of meat and meat products (61.2% vs. 40.1%), vegetables (77.7% vs. 58.3%), fish (93.2% vs. 79.8%), and dairy products (50.5% vs. 37.8%) than participants without CD, which indicates a higher consumption of non-traditional Mediterranean foods, such as meat and dairy products, and lower consumption of traditional Mediterranean, such as vegetables and fish, in participants with CD compared to participants without CD.

### 3.3. Ultra-Processed Foods’ Consumption and Adherence to the Mediterranean Diet

Figure 2 shows the Medi-Lite Score according to the level of UPF consumption in the diet (1st tertile ≤ 10%; 2nd tertile 10–17%; 3rd tertile ≥ 17%) in adults with and without CD, adjusted for age, gender, BMI, education level, civil status, and daily food consumption (g/day). Both participants with and without CD who had a higher intake of UPF reported a lower adherence to MD [9.0 (8.3–9.7) and 9.6 (9.2–10.1), respectively] than participants with medium [9.3 (8.6–10.1) and 10.7 (10.3–11.2), respectively] or low [9.9 (9.3–10.6) and 11.0 (10.6–11.5), respectively] consumption of UPF. This trend was statistically significant only in participants without CD (*p*<0.001).

## 4. Discussion

The present is the first study to assessed UPF consumption in relation to MD adherence in a group of adults with CD, in comparison to adults without CD, in Italy. In our study, UPF consumption did not differ according to the presence of CD. However, subjects with CD reported higher consumption of precooked pasta and pre-packaged breads than subjects without CD, with cereal-based products and sweets being the most common UPF in the diets of adults with CD. On the other hand, participants with CD had significantly lower MD adherence compared to participants without CD, with higher consumption of non-traditional Mediterranean foods, such as meat and dairy products, and lower consumption of traditional foods, such as vegetables and fish. Furthermore, an inverse trend was found between UPF consumption and MD adherence in adults with CD, although not statistically significant.

CD is a chronic autoimmune disease of the small intestine, for which the only current treatment is to follow a gluten-free diet [21]. Recent studies have reported a poorer nutritional quality associated with the gluten-free diet due to higher total fat and saturated fat contents and lower fiber content [16,22]. One of the possible causes has been attributed to the quality of gluten-free products, which are generally associated with processed products with low nutritional density [5,6,7]. Using the validated NFFQ questionnaire specifically designed to assess UPF consumption based on the NOVA classification, we analyzed UPF consumption in a group of adults with CD compared to a group of adults without CD. The higher consumption of ultra-processed pasta and bread we found in participants with CD is not surprising, as these are cereal-based products that naturally contain gluten. In fact, the UPF most consumed by the participants with CD were found to be agglutinated versions of cereal-based products and sweets, including snacks and biscuits, commonly known as gluten-free products. Various studies have shown the importance of these products, not only for their contribution to daily energy intake, but also as a source of dietary carbohydrates for patients with CD. In children and adolescents with CD, gluten-free products contributed between 24 and 36% of the total daily caloric intake and provided 49.5–73% of the total carbohydrate intake [6,16,23]. Although gluten-free products are clearly an important part of UPF consumption among people with CD, the total UPF consumption in our study did not differ from the intake of adults without CD. One possible hypothesis could be related to the use of gluten as an additive ingredient in UPF to improve their organoleptic properties [8], which limits the choice of UPF products among people with CD. For example, the use of different additives in the production of processed meat products [24] can explain the fact that the consumption of ultra-processed meat and fish-based foods in our study was lower in participants with CD than in participants without CD. Nevertheless, in contrast to our findings, a previous study conducted in Spain reported a higher UPF consumption in subjects with CD compared to subjects without CD. These differences can be determinate by the different methods used to evaluate UPF consumption, as well as the different age of participants. In fact, the research conducted in Spain included in the study children and adolescents, who tend to have a higher UPF consumption than adults [25], even in people with CD [26].

The gluten-free diet, despite being a restrictive dietary profile due to the total elimination of gluten-containing foods, should be a varied and balanced diet based on a healthy eating pattern [1]. The countries bordering the Mediterranean Sea are characterized by following the lifestyle promoted by the MD, a dietary pattern associated with robust scientific evidence on its health benefits [13]. MD is characterized by plant-based foods, with high consumption of vegetables, fruits, whole grains, legumes, nuts, and seeds; olive oil as the main source of fat; moderate consumption of fish, dairy products, and eggs; and low consumption of meat. All the food groups mentioned above could be included in a gluten-free diet, with only the exception of whole grains, which should be substituted by naturally gluten-free cereals, such as rice, corn, millet, or teff. For this reason, MD could also be a suitable dietary pattern for people with CD. However, in line with the study published by Morreale et al., [12] our study shows significantly lower MD adherence in adults with CD compared to adults without CD. Specifically, adults with CD reported a higher consumption of non-traditional Mediterranean foods, such as meat, milk, and dairy products, and a lower consumption of traditional foods, such as vegetables and fish. In both studies, healthy adults also reported low values for the consumption of typical Mediterranean foods, highlighting the decrease in MD adherence already observed in several studies [27,28,29].

As the current literature suggests, the change in eating habits due to the introduction of UPF may replace the consumption of fresh foods that form the basis of traditional diets [30]. In a previous study conducted on Italian adults, UPF consumption was indeed associated with lower adherence to MD [31]. Data on subjects with CD, however, are more limited. To date, only a Spanish study conducted in children and adolescents has analyzed this, finding that patients with CD who consumed higher energy intake from UPF had lower adherence to MD and vice versa [14]. In our study, we also observed a relationship between higher UPF consumption and lower adherence to MD, although the result reached statistical significance only in controls. This could be due to the fact that the case population was limited and underlines the need for further studies on this topic. In fact, due to the increased risk of non-communicable diseases associated with high UPF consumption, such as cardiovascular diseases, cerebrovascular disease, and depression [32], it is of great importance to promote a healthy dietary pattern, such as MD and a limitation of UPF. Furthermore, considering the presence of gluten-free products as a source of UPF in the diets of people with CD, it should also be relevant to promote the consumption of naturally gluten-free cereals to improve the diet quality of people with CD. In fact, some studies have already demonstrated the efficacy of nutrition education in people with CD, reporting an improvement in UPF consumption and better adherence to MD in children, adolescents [15,33], and adults [26].

The present study has some limitations that need to be underlined. Firstly, due to its design, we cannot establish any cause–effect relationship. Secondly, we cannot exclude a sample bias where the participants with and without CD selected for our study may not accurately represent the members of the population. The use of a self-administered online survey is also a limitation of this study, because although it is a very useful and low-cost tool, at the same time, it may lead to recall bias and misclassification errors. However, we used a validated questionnaire specifically designed to estimate UPF intake according to NOVA classification, thus avoiding the misclassification of foods according to the extent and purpose of food processing. Moreover, the evaluation of food intake was made with the proportions, by weight, of UPF in the diet. This method is more suitable than using the energy proportion, to avoid underestimating non-caloric beverages, such as artificially sweetened beverages, as well as various of the non-nutritional components related to food processing, such as additives, neoformed contaminants, food contact materials, or alterations of the food matrix.

## 5. Conclusions

In conclusion, our study suggests higher consumption of ultra-processed pasta and bread-based foods and lower MD adherence in adults with CD compared to adults without CD. In addition, higher UPF consumption was observed in adults with CD with lower adherence to MD. These findings highlight the importance of improving the nutrition education of people with CD, which should not only focus on gluten exclusion, but also on promoting a balanced and healthy dietary pattern. Special attention should be paid to the intake of gluten-free products, while encouraging the consumption of naturally gluten-free cereal products in order to limit UPF consumption by people with CD.

## Figures and Tables

**Figure 1 nutrients-15-00938-f001:**
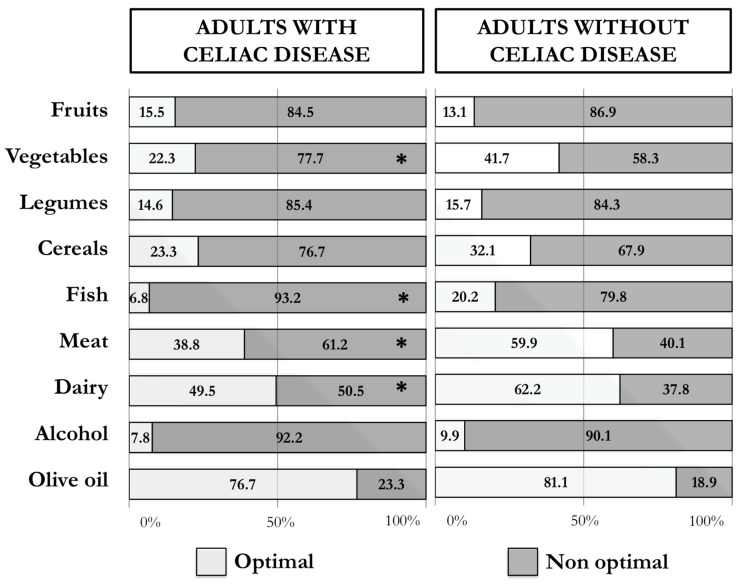
Percentages of adults with and without celiac disease who reported an optimal (2 points) and non-optimal (≤1 points) intake of the single food groups of the Medi-Lite score (* *p*-value < 0.05).

**Figure 2 nutrients-15-00938-f002:**
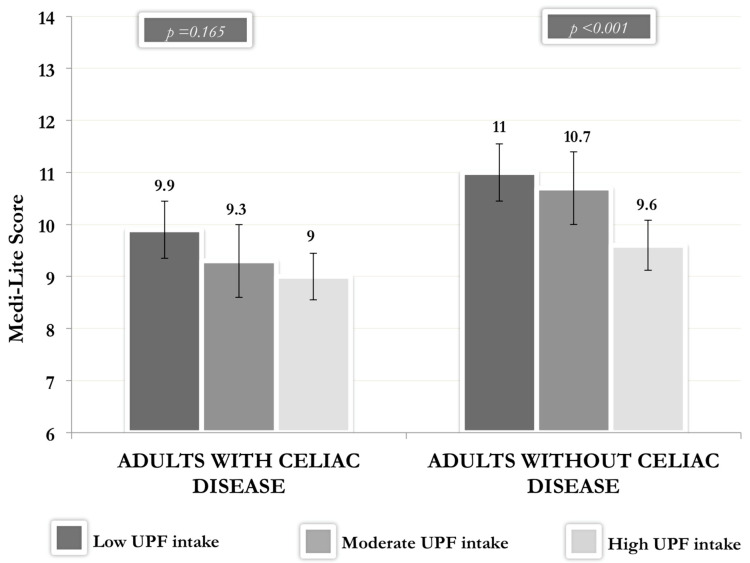
Medi-Lite Score according to the level of UPF consumption in adults with and without celiac disease. Legend: UPF: ultra-processed foods. Vertical bars represent the 95% confidence interval (CI).

**Table 1 nutrients-15-00938-t001:** Sociodemographic characteristics of the study population.

	Adults with Celiac Disease (*n* = 103)	Adults without Celiac Disease (*n* = 312)	*p*-Value
Age (years)	40.4 ± 12.6	39.3 ± 13.3	0.471
Gender, *n* (% women)	89 (86.4)	261 (83.7)	0.505
Body weight (kg)	63.4 ± 12.1	64.8 ± 14	0.823
BMI (kg/m^2^)	23.2 ± 4.2	23.1 ± 4.1	0.740
BMI ≥ 25, *n* (%)	25 (24)	78 (25.0)	0.882
Education level			
Secondary school, *n* (%)	10 (9.7)	14 (4.5)	0.049
High school, *n* (%)	36 (35)	133 (42.6)	0.169
University degree, *n* (%)	57 (55.3)	165 (52.9)	0.206
Civil status			
Single, *n* (%)	37 (35.9)	131 (42)	0.276
Married/partner, *n* (%)	61 (59.2)	150 (48.1)	0.050
Divorced/Widowed, *n* (%)	5 (4.9)	31(10)	0.112
Years of disease	11.4 ± 8.6	-	-

Legend: BMI = Body Mass Index. Data are reported as mean ± standard deviation (SD) or number and percentage (%), as appropriate.

**Table 2 nutrients-15-00938-t002:** UPF intake (g/day) in adults according to the presence of celiac disease.

	Adults with Celiac Disease (*n* = 103)	Adults without Celiac Disease (*n* = 312)	*p*-Value
**Vegetables and legumes UPF**	15.9 (5.4–26.4)	30.1 (24.1–36.2)	0.021
Ready-to-heat vegetables and legumes (with added ingredients)	15.9 (5.4–26.4)	30.1 (24.1–36.2)	0.021
**Cereals and tubers UPF**	66.5 (57.0–75.9)	56.2 (50.8–61.6)	0.066
Ready-to-heat pasta/gnocchi dishes	9.7 (7.2–12.1)	6.6 (5.1–8.0)	0.033
Pre-packaged breads and bread alternatives	26.3 (21.7–30.9)	19.6 (17.0–22.2)	0.012
Pre-packaged pizza, focaccia, sandwich, and savory pies	15.9 (11.6–20.2)	13.0 (10.5–15.5)	0.254
Breakfast cereals and energy bars (with added sugar)	5.1 (3.0–7.1)	5.6 (4.4–6.8)	0.667
Pre-packaged potatoes, croquets, and instant soups	9.2 (6.0–12.3)	11.1 (9.3–13.0)	0.292
**Meat and fish UPF**	8.7 (6.1–11.2)	12.2 (10.8–13.8)	0.016
Nuggets, sticks, sausages, burgers, and other reconstituted meat products	7.5 (5.3–9.8)	10.2 (8.9–11.5)	0.043
Fish nuggets, fish sticks, and other reconstituted fish products	1.1 (0.3–1.9)	2.1 (1.6–2.5)	0.043
**Milk and dairy products UPF**	28.6 (19.9–37.3)	33.4 (28.4–38.4)	0.346
Milk beverages (e.g., probiotic milk with added sugar)	6.4 (1.7–11.1)	7.9 (5.2–10.6)	0.579
Fruit or flavored yogurts (e.g., vanilla flavored)	20.6 (13.4–27.8)	23.9 (19.8–28.1)	0.431
Melted cheese (also used as a sandwich filling)	1.6 (1.0–2.2)	1.6 (1.2–1.9)	0.949
**Fats and seasoning UPF**	4.9 (3.5–6.2)	5.2 (4.5–6.0)	0.654
Margarines and other spreads, or instant sauces (e.g., mayonnaise, ketchup)	4.9 (3.5–6.2)	5.2 (4.5–6.0)	0.654
**Sweets and Sweeteners UPF**	51.4 (44.4–58.4)	45.8 (41.7–49.8)	0.173
Pre-packaged biscuits, cakes, snacks, and ice-cream	38.4 (32.1–44.7)	33.3 (29.7–36.9)	0.169
Chocolate, spreads (e.g., nut spread), and candies	13.0 (10.2–15.8)	12.4 (10.8–14.0)	0.740
**Beverages UPF**	36.9 (24.9–49.0)	46.3 (39.4–53.2)	0.186
Alcoholic beverages, soft and energy drinks (e.g., iced tea, coke)	36.9 (24.9–49.0)	46.3 (39.4–53.2)	0.186
**Other UPF**	26.6 (17.3–35.8)	19.9 (14.6–25.1)	0.216
Plant-based dairy substitutes (e.g., soy drinks)	25.0 (16.3–33.7)	16.9 (11.9–21.9)	0.112
Plant-based meat substitutes (e.g., veggie burger)	1.6 (0.2–3.0)	3.0 (2.2–3.8)	0.078

Legend: UPF = ultra-processed foods. Data are reported as geometric mean and 95% confidence interval (CI). General linear model adjusted for age, gender, BMI, education level, civil status, and daily food consumption (g/day).

## Data Availability

Additional information is available on request from the corresponding author.

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
