# Peer review of "Adherence to the Mediterranean Diet and Ultra-Processed Foods Consumption in a Group of Italian Patients with Celiac Disease"

_nutrients, 2023, doi:10.3390/nu15040938_

Round 1

Reviewer 1 Report

The present study is very interesting, since it investigates the relation between ultra processed food intake and Mediterranean diet in celiac and non-celiac adults. This study presents high scientific quality and relevance to the field of this journal.

The study and analysys were well conducted.  The results were well discussed too.

 Some minor concerns

 METHODS

I have one doubt: “To assess the possible relationship between MD adherence and UPF consumption in coeliac and non-coeliac participants, they were divided into tertiles (1st tertile ≤ 10%; 117 2nd tertile = 10-17%; 3rd tertile ≥ 17%) according to their level of UPF intake”.

Why did the authors use this cut off (1st tertile ≤ 10%; 117 2nd tertile = 10-17%; 3rd tertile ≥ 17%) . Are there any reference  or reason for this value Or was it arbitrary?

Is the percentage of UPF intake (10%, 10-17% and >17%) in relation to grams or kcal? Please specify this information in the text.

DISCUSSION

Line 201: “which are generally associated with processed products with low  nutritional density [5-7].” Would be more adequate the term ultra-processed food?

Lines 204-205 “In 204 our study, coeliac subjects reported a higher consumption of precooked pasta and 205 pre-packaged bread than non-coeliac subjects” This information was also described in the first paragraph of discussion.

Reviewer 2 Report

Please see attached document with my review
